# Intraoperative Hypothermia Induces Vascular Dysfunction in the CA1 Region of Rat Hippocampus

**DOI:** 10.3390/brainsci12060692

**Published:** 2022-05-27

**Authors:** Tianjia Li, Guangyan Xu, Jie Yi, Yuguang Huang

**Affiliations:** Department of Anesthesiology, Peking Union Medical College Hospital, Chinese Academy of Medical Sciences and Peking Union Medical College, Beijing 100730, China; litianjia@pumch.cn (T.L.); bonnie_xgy08@163.com (G.X.); garypumch@163.com (Y.H.)

**Keywords:** intraoperative hypothermia, hippocampus, CA1 region, vascular endothelial cells, endothelial dysfunction, vascular smooth muscle cells, phenotypic transformation, activity-regulated cytoskeleton-associated protein, reactive oxygen species, endothelial nitric oxide synthase

## Abstract

Intraoperative hypothermia is very common and leads to memory decline. The hippocampus is responsible for memory formation. As a functional core area, the cornu ammonis 1 (CA1) region of the hippocampus contains abundant blood vessels and is susceptible to ischemia. The aim of the study was to explore vascular function and neuronal state in the CA1 region of rats undergoing intraoperative hypothermia. The neuronal morphological change and activity-regulated cytoskeleton-associated protein (Arc) expression were evaluated by haematoxylin-eosin staining and immunofluorescence respectively. Histology and immunohistochemistry were used to assess vascular function. Results showed that intraoperative hypothermia inhibited the expression of vascular endothelial growth factor and endothelial nitric oxide synthase, and caused reactive oxygen species accumulation. Additionally, the phenotype of vascular smooth muscle cells was transformed from contractile to synthetic, showing a decrease in smooth muscle myosin heavy chain and an increase in osteopontin. Ultimately, vascular dysfunction caused neuronal pyknosis in the CA1 region and reduced memory-related Arc expression. In conclusion, neuronal disorder in the CA1 region was caused by intraoperative hypothermia-related vascular dysfunction. This study could provide a novel understanding of the effect of intraoperative hypothermia in the hippocampus, which might identify a new research target and treatment strategy.

## 1. Introduction

With an increase in longevity and a growing older population, many people may undergo operations at least once in their lifetime. Intraoperative hypothermia, defined as a core temperature <36 °C, is a common complication, and its incidence can be up to 25–70% [1,2]. Previous studies and our epidemiological investigation found that intraoperative hypothermia postponed memory recovery, and increased dependency on families and society [3,4]. However, the underlying mechanism of this phenomenon remained unclear. Recently, studies showed that there were no obvious brain abnormalities related to synaptic plasticity, neurotransmitters, or other physiological function [5,6,7,8,9,10].

Brain tissue is sensitive to ischemia, as cerebral function is critically dependent on adequate blood supply [11]. Despite representing only 2% of total body weight, the brain receives over 15% of the cardiac output and consumes more than 20% of the glucose and oxygen sources [12]. The vascular contributions to cognitive and memory decline should be one of the priority research areas for potential breakthroughs [13]. Studies on Alzheimer’s and other memory disorders show that the hippocampus is vulnerable to hypoperfusion and vascular density, diameter, and branch angle are reduced in the hippocampus [13,14]. Moreover, as a crucial structure of neural circuits in contextual retrieval and emotional learning, the hippocampus functionally binds with the prefrontal cortex, suggesting that it is preferentially vulnerable [15].

The hippocampus is a crucial structure for connecting with other brain structures and storing memories which are embedded deep into the temporal lobe [16]. This hippocampal region needs a large blood supply to function properly supplied by the posterior cerebral artery and the anterior choroidal artery [17], which branch into numerous arterioles passing through the hippocampus [18]. In memory decline diseases, vascular dysfunction is often the early features [12,19]. Patients with vascular diseases, such as atherosclerosis, are likely more susceptible to memory impairment [20]. Thus, vascular dysfunction may be the mechanism of the intraoperative hypothermia-related hippocampal damage.

Blood vessels are composed of vascular endothelial cells (VECs) and vascular smooth muscle cells (VSMCs). For VECs, endothelial nitric oxide synthase (eNOS) is the key functional molecule that catalyzes nitric oxide to dilate blood vessels and ensure blood supply [21]. Vascular endothelial growth factor (VEGF) is responsible for the survival of VECs, which is required for angiogenesis and the maintenance of endothelium integrity [22,23]. Functionally, VSMCs can be divided into contractile and synthetic phenotypes according to different characteristics [24,25]. Their phenotypic transformation is involved in the pathophysiology. Physiologically, marked by the presence of smooth muscle myosin heavy chain (SMMHC), VSMCs are in the state of contractile phenotype, which regulates vascular tension to control blood supply. In response to pathological stimulation, VSMCs are transformed to the synthetic phenotype with the expression of osteopontin. The phenotypic transformation induces VSMCs proliferation and migration, eventually leading to the overexpression of retinol-binding protein (RBP) and stenosis of the vascular lumen [26,27].

VECs, VSMCs, cerebrovascular basement membrane, pericytes, astrocytes and neurons are collectively referred to as the neurovascular unit (NVU) [28]. Physiologically, the interaction between vascular function and neuronal activity is known as neurovascular coupling [12,29,30]. This mechanism ensures the delivery of oxygen and nutrients in the vascular network of arteries, arterioles, and capillaries. And then oxygen and nutrients are transport across the blood-brain barrier [31], that composes of endothelial cells, pericytes, and astrocytic end feet and forms a protective sheath around brain capillaries [32]. Regarding the NVU and the hypothesis that neurovascular coupling, VSMC contractility, and endothelial-related arteriolar dilation can become dysfunctional in memory decline [12], our study focused on the arterioles connecting neurovascular dysfunction and neurodegeneration in the hippocampus.

Cytoarchitecturally, the rat hippocampus is divided into cornu ammonis 1 (CA1), CA3, and dentate gyrus (DG) regions. The head of the hippocampus is mainly composed of the CA1 region with a complete signal output loop [33,34]. As a functional core area, the CA1 region is particularly sensitive to ischemia [35,36]. Moreover, enduring forms of memory rely on the structural connectivity of synapses that typically require activity-dependent expression of the activity-regulated cytoskeleton-associated protein (Arc). In response to external information, Arc is the immediate-early protein that converts synaptic input into neuronal changes and forms memory [37,38]. Arc knockout mice fail to form long-lasting memories. Moreover, reducing basal Arc expression by less than 60% can significantly attenuate spatial memory [39]. Because of these strong correlations it has been widely assumed that Arc plays a role in memory formation and indicates hippocampal function.

The inadvertent intraoperative hypothermia has a high incidence, and the memory decline is one of its important complications. Few studies addressed the effect of intraoperative hypothermia in the memory decline. As the functional core area of memory, the hippocampal CA1 region is rich in blood vessels and susceptible to ischemia. Therefore, the mechanism of ischemic injury of the CA1 region might provide a new research target and the development of a prevention strategy for intraoperative hypothermia-induced memory decline. The purpose of this study was to explore the vascular function in the CA1 region and to analyze the memory-related neuron status.

## 2. Materials and Methods

### 2.1. Ethics Statement

This study complied with the Guide for the Care and Use of Laboratory Animals by the National Institutes of Health and the Animal Management Rule of the Chinese Ministry of Health, and was approved by the Animal Care Committee of Peking Union Medical College (XHDW-2019-003).

### 2.2. Animals and Experimental Design

Adult male Sprague Dawley rats (300–400 g, 12-weeks-old) were purchased from HFK Bioscience Co., Ltd. (Beijing, China). All rats were housed in a specific pathogen-free room (temperature 23 ± 1 °C, 12 h light/dark cycle) with free access to food and water. These rats were randomly divided into three groups (*n* = 10 per group). The intraoperative hypothermia group received exploratory laparotomies on cryogenic pads (rectal temperature 33 ± 0.5 °C). The intraoperative normothermia group received exploratory laparotomies on temperature blankets (rectal temperature 37 ± 0.5 °C). The surgical procedure lasted for 60 minutes. The control group was fed normally without operation (rectal temperature 37 ± 0.5 °C). After 12-days, the rats were anesthetized by the injection of pentobarbital sodium to collect hippocampus for following experiments. Each harvested hippocampus was evenly divided into two parts, and was fixed in paraffin and frozen in liquid nitrogen respectively.

During operation, an abdominal median incision of approximately 3 cm was made to allow penetration of the peritoneal cavity, and 5–10 cm of the small intestine was exteriorised and left in air for 60 min. The intestine was then placed inside the peritoneal cavity, and the wound was sutured with 4-0 non-absorbable sutures in three layers consisting of the peritoneal lining, abdominal muscles, and skin. Core body temperatures were measured using a rectal temperature probe. After the wounds were sutured, the rats were immediately intraperitoneally injected with penicillin (60,000 U) to prevent postoperative infections, and the body temperatures of those in intraoperative hypothermia group were raised to 37 ± 0.5 °C using a heating pad.

### 2.3. Histology and Immunohistochemistry

The harvested hippocampi (*n* = 10 per group) were fixed in 4% paraformaldehyde, embedded in paraffin. As for the orientation and distribution of vasculature in hippocampus, the vascular network exhibited a rake-like distribution of equally spaced transverse hippocampal arterioles arising from longitudinal hippocampal arteriole. The capillaries mainly formed the microvascular network in the distal end. And the hippocampal arterioles entered the hippocampus perpendicularly to the coronal plane [13]. In our study, the hippocampal paraffin block was sectioned along the coronal plane (5-μm thick) to ensure the hippocampal vasculature was properly outlined [29,40,41]. Additionally, quantitative data on the vasculature of the hippocampus of mice showed the maximum vascular diameter (30.02 ± 0.90 μm) [13]. For the function of NVU, our study focused on the arterioles of hippocampus, and the mean diameter of arterioles in the study was 21.16 ± 0.87 μm (Carl Zeiss Jena, Oberkochen, Germany).

The slides were stained with hematoxylin-eosin (H&E), and observed by microscopy (Ti-S, Olympus FluoView software, Olympus, Japan). The total number of damaged neurons in each image was counted and analysed by Image-Pro Plus 6.0 Software. Immunohistochemical (IHC) staining was also performed to assess the expression of vascular functional molecules. After blocking in phosphate-buffered saline containing 5% goat serum for 30 min at room temperature (20–25 °C), the sections were incubated overnight at 4 °C with primary antibodies. The following primary antibodies were used: anti-CD31 (1:2000, #ab182981, Abcam, Cambridge, UK), anti-eNOS (1:2000, #ab76198, Abcam), anti-VEGF (1:500, #19003-1-AP, Proteintech Group, Chicago, IL, USA), anti-SMMHC (1:200, #21404-1-AP, Proteintech Group), anti-osteopontin (1:200, #22952-1-AP, Proteintech Group), and anti-RBP (1:200, #22683-1-AP, Proteintech Group). The following day, signal amplification was performed with horseradish peroxidase-conjugated goat anti-rabbit or anti-mouse IgG secondary antibody (1:500; ZhongshanJinqiao Biotechnology Co., Ltd., Beijing, China). Cytokines expression levels were quantified using integrated optical density (IOD) values generated by Image-Pro Plus 6.0 software.

### 2.4. Detection of Reactive Oxygen Species (ROS)

Dihydroethidium was used to evaluate ROS formation (*n* = 10 per group) in the frozen sections of hippocampus (12 μm thick). The sections were incubated with dihydroethidium (10 μmol/L; Sigma-Aldrich, St Louis, MO, USA) for 30 min at 37 °C in the dark, and then washed with phosphate-buffered saline three times. The nuclei were stained with 4′,6-diamidino-2-phenylindole (DAPI). The level of ROS was assessed by fluorescence microscopy. Confocal images of hippocampal sections were sequentially acquired with Zeiss AIM software on a Zeiss LSM 700 confocal microscope system (Carl Zeiss Jena, Oberkochen, Germany). In fluorescence intensity analysis, the whole section was scanned to one image and stored as the “MView MRXS File”. And then we used the CaseViewer 2.4 software (3DHISTECH Ltd., Budapest, Hungary) to get multiple images from different regions.

### 2.5. Immunofluorescence

Immunofluorescence staining was performed on the sectioned hippocampus (*n* = 10 per group) to assess the expression of Arc. Hippocampus sections were blocked with 5% goat serum for 30 min at room temperature, then incubated with primary anti-Arc (sc-17839, 1:100, Santa Cruz, CA, USA) and anti-NeuN (1:100, ab12763, Abcam) overnight at 4 °C. The secondary antibodies used were anti-mouse and anti-rabbit respectively (1:1000; Invitrogen, Carlsbad, CA, USA). Sections were incubated with the secondary antibodies in the dark for 2 h at room temperature. The nuclei were then stained with DAPI. The anti-NeuN was used as neuronal marker. The expression level of Arc was assessed by fluorescence microscopy. Confocal images of sections were sequentially acquired with Zeiss AIM software on a Zeiss LSM 700 confocal microscope system. The analysis method is the same as ROS mentioned above.

### 2.6. Statistics

All data were expressed as mean ± SEM and were compared by ANOVA and Tukey post hoc test (using SPSS26.0). Statistical significance was determined at *p* < 0.05.

## 3. Results

### 3.1. The Abundance of Blood Vessels in the CA1 Region of the Hippocampus and Its Relationship with Hippocampal Function

CD31 is the endothelial marker of vascular tissue. There are abundant blood vessels in the hippocampus (Figure 1A), indicating a large demand for blood supply. Compared with other regions, there are more blood vessels in the CA1 region (Figure 1B), which demonstrates its susceptibility to ischemia. Additionally, the schematic diagram displays the development from vascular dysfunction to neuronal damage (Figure 1C). First, vascular dysfunction consists of VECs dysfunction and VSMCs overproliferation. Second, the endothelial dysfunction is characterized by the low expression of VEGF and eNOS, while the excessive proliferation of VSMCs is induced by the phenotypic transformation. VEGF is the required growth factor for the survival of VECs and the maintenance of endothelium integrity. eNOS is the functional molecule that synthesizes nitric oxide and maintains vascular diastolic function. For VSMCs, the phenotypic transformation from physiological contraction to pathological synthesis indicated an excessive proliferation and subsequent vascular stenosis, marked by osteopontin overexpression. Moreover, the low-expression of eNOS can cause ROS accumulation, further promoting the phenotypic transformation. All these vascular abnormalities eventually led to insufficient blood supply, causing hippocampal neuronal damage, as indicated by neuronal pyknosis and the low-expression of Arc. As hippocampus is the key region for memory, vascular dysfunction ultimately causes memory decline. Briefly, the schematic diagram in the Figure 1 provides a general description of the research.

### 3.2. Intraoperative Hypothermia Caused Dysfunction of VECs and Phenotypic Transformation of VSMCs

In representative cross-sections of the hippocampus, black rectangles in the CA1 region were magnified to 20× to show blood vessels (Figure 2). For VECs, intraoperative hypothermia significantly inhibited the expression of VEGF and eNOS (Figure 2A–D), while intraoperative normothermia preserved those expressions. As VEGF and eNOS function to preserve endothelial integrity and reduce oxidative damage, respectively, ROS was accumulated in the intraoperative hypothermia group (Figure 2E,F), indicating the presence of hippocampal oxidative damage. For VSMCs, intraoperative hypothermia led to the phenotypic transformation, in which the expression of contractile phenotype marker SMMHC was decreased and the synthetic phenotype marker osteopontin was increased (Figure 2G–J). Inversely, intraoperative normothermia maintained the physiological contractile phenotype. The dysfunction of VECs and phenotypic transformation of VSMCs could induce atherosclerosis, which displayed as the overexpression of the proliferation regulator RBP in the intraoperative hypothermia group (Figure 2K,L). These results demonstrated that intraoperative hypothermia disturbed vascular function of the CA1 region, leading to subsequent neuronal damage.

### 3.3. The Overall Impact of Intraoperative Hypothermia on the Blood Vessel

The schematic diagram shows pathological changes in blood vessels induced by intraoperative hypothermia (Figure 3). For VECs, the dysfunction was characterized by disturbed organelles with decreased expression of VEGF and eNOS. In endothelium, increased vascular permeability could lead to infiltration of inflammatory cells and thrombosis. Additionally, ROS accumulated in the blood vessels, compromising vascular integrity. For VSMCs, actin is the main intracellular protein in the physiological contractile phenotype with vasomotor function. After the phenotypic transformation induced by intraoperative hypothermia, VSMCs deformed from shuttle to oval in shape. In microstructure, VSMCs could synthesize more protein and possessed more organelles, such as endoplasmic reticulum, Golgi apparatus, etc. Ultimately, the synthetic phenotype of highly proliferative VSMCs caused vascular stenosis and hypoperfusion.

### 3.4. Intraoperative Hypothermia Damaged Hippocampal Neurons in the CA1 Region

Figure 4A illustrates three regions of the rat hippocampus: CA1, CA2, and DG. The CA1 region is most susceptible to ischemia. H&E staining was used to analyze the state of hippocampal neurons (Figure 4B,C). The CA1 region was framed in the 4× magnification image. In 20× magnification images, there were neuronal pyknosis and necrosis in the intraoperative hypothermia group. In contrast, in control and intraoperative normothermia groups, neurons were round or oval, with lightly stained cytoplasm. These morphological changes of neurons in the CA1 region demonstrated that intraoperative hypothermia caused neuronal damage. Besides, the neuronal activity was marked by Arc expression, which plays a critical role in memory storage. The hippocampal immunofluorescence analysis showed that the expression of Arc in the CA1 region was significantly decreased by intraoperative hypothermia (Figure 4D,E). Together with previous results, the study indicated intraoperative hypothermia-related vascular dysfunction induced hippocampal neuronal damage, especially in the CA1 region with high susceptibility to ischemia.

## 4. Discussion

Clinical investigations and rat maze experiments both demonstrated that intraoperative hypothermia could lead to memory decline [4,42]. As memory is generated in the hippocampus, intraoperative hypothermia may negatively impact the hippocampus. While blood supply is critically required for neuronal function, hippocampal blood vessels are very thin and lack capillary anastomosis (Figure 1). In the high-resolution mapping of brain vasculature, the low vascular diameter, length density (total vascular length per tissue volume) and volume fraction (total vascular volume relative to the total tissue volume) of the hippocampus also suggest that the hippocampus is more vulnerable to cerebrovascular dysfunction [13]. Therefore, the hippocampus is particularly sensitive to ischemia. Studies of Alzheimer’s disease also revealed ischemic damage in the hippocampus [14]. Thus, the vascular function could be a key point to explore the effect of intraoperative hypothermia on hippocampus. The hippocampus in a rat is divided into CA1, CA3, and DG regions. The CA1 region, with widespread projections, is the key signal output node of the hippocampal memory circuit [43]. Due to its functional features, the CA1 region needs more blood supply and is highly susceptible to ischemia.

Intraoperative hypothermia induced vascular dysfunction in the CA1 region of rat hippocampus. The vascular wall consists of VECs and VSMCs, which are important in preserving vascular function. VECs line the vascular lumen and contact with blood. Their dysfunction is the pathological basis of thrombosis and stenosis. Among them, VEGF is required for the survival of VECs and angiogenesis. Intraoperative hypothermia significantly reduced VEGF expression (Figure 2A). The decreased VEGF could result in shedding of VECs and changes in vascular permeability that attenuated blood supply. VEGF also has a neurotrophic effect [22], especially in the ischemia-sensitive CA1 region. Additionally, eNOS induces nitric oxide that preserves vascular relaxation and eliminates ROS. In vivo measurements, nitric oxide increased the local cerebral blood flow in hippocampus, while ROS compromised nitric oxide mediated neurovascular response during memory decline [44]. eNOS expression was significantly decreased in the intraoperative hypothermia group in comparison to that of the control group (Figure 2C). The decrease in eNOS indicated the presence of endothelial impairment, which could reduce hippocampal blood supply. Furthermore, the decrease in antioxidant molecule eNOS and subsequent reduction in blood supply resulted in the accumulation of ROS, which induced neuronal oxidative damage in the intraoperative hypothermia group (Figure 2E). In clinical practice, the factors contributing to the inadvertent intraoperative hypothermia might also lead to the vascular dysfunction and the accumulation of ROS, such as impairment of thermoregulation by general anesthesia and undergoing major-plus surgery [45,46]. Collectively, these results showed that intraoperative hypothermia disturbed the function of VECs, subsequently reducing the blood supply of the hippocampal CA1 region.

VSMCs constitute the media of the vascular wall and control the vasomotor to regulate blood supply. Normal vascular structure and function depend on the contractile phenotype of VSMCs with the presence of SMMHC that is an actin-based motor protein essential to VSMCs motility. Intraoperative hypothermia caused the SMMHC suppression and osteopontin overexpression (Figure 2G–J). The phenotypic transformation leads to wall thickening and decreased compliance. Osteopontin mediates cell-cell and cell-extracellular matrix interactions, which is associated with VSMCs adhesion and vascular remodeling [47]. Overall, these pathological changes can lead to the hypoperfusion in hippocampus. With the regard to the NVU, neuron also can induce nitric oxide and regulate VSMCs relaxation [12,48]. Further neural signal pathway studies shown that activation of neuronal N-methyl D-aspartate receptor could lead to concentration dependent nitric oxide-mediated dilatation of arterioles [49,50]. Therefore, the intraoperative hypothermia-induced vascular dysfunction might also involve the N-methyl D-aspartate receptor signal pathway which will be detected in our following study. Additionally, more organelles, such as the endoplasmic reticulum and Golgi complex, could be observed in synthetic VSMCs (Figure 3), indicating the enhancement in synthesis and secretory functions. Besides, the phenotypic transformation could lead to angiosclerosis [51]. Overexpression of RBP was shown in the intraoperative hypothermia group (Figure 2K), which is implicated in lipid deposition and mineralization. The series of pathological changes in the contractile phenotype exacerbates luminal stenosis and hippocampus hypoperfusion. Within the NVU, the cellular composition differs along the vascular tree [12]. Pericytes in the terminal capillary level are the ones with contractile properties and control blood flow, as shown for VSMCs in the upper arteriole segment of NVU [52]. Memory decline diseases also demonstrate pericyte dysfunction in the hippocampus [31,53]. Collectively, VSMCs were transformed into the synthetic phenotype by intraoperative hypothermia. The phenotypic transformation could result in vasomotor dysfunction and vascular stenosis, ultimately attenuating the blood supply of the CA1 region.

As two major components of blood vessels, VECs and VSMCs have complementary functions. The intact VECs act as a barrier to protect VSMCs from blood flow stimulation, which maintains the normal contractile function. In addition, endothelial release of nitric oxide leads to VSMC relaxation and arteriolar dilation. Following endothelial damage, the subsequent excessive proliferation of VSMCs could lead to atherosclerosis [54,55]. Moreover, the accumulation of ROS also induced the development of the synthetic phenotype in VSMCs [56]. Likewise, the synthetic phenotype VSMCs could excessively proliferate and destroy the integrity of the endodermis, which further exacerbates the progression of vascular disorder. Eventually, these series of events resulted in lumen narrowing and hippocampus hypoperfusion.

Hypoperfusion could damage structural integrity, such as gray matter atrophy and white matter microstructure disorder [57]. In the vulnerable CA1 region, neurons are more sensitive to ischemia and prone to pathological changes. There is clear neuronal pyknosis in the CA1 region of the intraoperative hypothermia group (Figure 4B). In terms of structure and function, the CA1 region is divided into the pyramidal, polymorphic, and molecular layers, of which individual values of neuronal density are relatively heterogeneous. The bodies of neurons are mainly located in the pyramidal layer, while axons and dendrites are in the polymorphic and molecular layers. Not all neuronal layers in the CA1 region have the same sensitivity to ischemia. We observed the neuronal pyknosis mainly in the pyramidal layer (Figure 4B). This finding confirms that the CA1 pyramidal neurons were sensitive to intraoperative hypothermia-related hypoperfusion. Additionally, the vascular dysfunction contributed to the activation of astrocytes and the ensuing inflammatory response [32], and neuronal damage could disturb specific neurochemical content. Particularly, the low expression of Arc further demonstrated neuronal damage (Figure 4D).

Arc is an active neuronal marker of hippocampal function. As the immediate-early protein, Arc accumulates in synapse and drives the formation of memory. Moreover, during the recall of memory, Arc has an anatomical distribution in the CA1 region. In this study, the expression of Arc in the CA1 region was decreased by intraoperative hypothermia (Figure 4). In addition, our maze experiment showed that reducing Arc expression in the CA1 region could attenuate the spatial memory of rats [42]. The low expression of Arc is associated with memory decline in different neurological diseases. Alzheimer’s disease shows a depletion of Arc in the hippocampus [5,58]. Arc knockout mice also failed to form long-term memories and would forget what they learned 24 h earlier [39]. In electrophysiology, decreased expression of Arc resulted in depolarization through the inhibitory neurotransmitter GABA and intracellular Cl- [59], which might be the underlying neurotransmitter mechanism of hippocampal neuronal damage in response to intraoperative hypothermia. The Arc reduction observed in this study further revealed the neuronal damage in the CA1 region.

The regulation of cerebral blood flow is essential for normal brain function. Specifically, the neurovascular coupling is modulated by NVU that interacts with local neuron activation [15,60,61,62]. Disrupted NVU and neurovascular uncoupling were found in the early stages of neurological disorders [12,63,64]. Our study showed NVU dysfuntion that VSMC contractility and endothelial-related arteriolar repairment became dysfunctional in the intraoperative hypothermia-induced memory decline.

As the anatomical and methodological difficulties [15,18], our study has a limitation in measuring the coordinated interaction between neurons and vasculature. Probing dynamically the hippocampal blood flow might allow characterization of vascular function during intraoperative hypothermia, which is what needs to be achieved in our future research. Besides, neuronal pyknosis (Figure 4b) implies that neurodegeneration can reflect the impairment and dysfunction of the mnemonic and cognitive nervous systems [65]. Except for the dynamic blood flow detection, magnetic resonance imaging and positron emission tomography may help identify the affected region of hippocampal neuron pyknosis [15,31,66]. Furthermore, during the progression of neurodegeneration, such as Alzheimer’s disease and Parkinson’s disease, neuroinflammation is the common prelude [67]. Vascular dysfunction induces hypoperfusion, increase of ROS, and oligodendrocyte malfunction, leading to the release of inflammatory cytokines, such as TNF-α, IL-1β, IL-6, and IL-10, which amplifies the risk of vascular dysfunction [67]. The alterations of inflammatory factors levels were observed in the cerebrospinal fluid of neurodegeneration diseases [68]. The inflammatory factors detection in cerebrospinal fluid may further reveal the mystery. Additionally, studies show that the therapeutic hypothermia is a neuroprotective strategy, and the mechanisms involved are complex, such as reducing oxygen requirement and cytotoxic edema, interrupting anoxic injury, preventing free-radical injury, protecting the blood-brain barrier integrity, and etc. [69,70,71,72]. In other words, the precise protective mechanisms are not yet fully understood. As for our study, the memory decline was observed after the inadvertent intraoperative hypothermia both in clinical practice and animal experiment [4,42,45,46]. In regard to cerebral blood flow, the therapeutic hypothermia also induced reduction in post-ischemic hyperperfusion and sustained hypoperfusion [72]. Anyway, there are differences between therapeutic hypothermia and inadvertent intraoperative hypothermia in their neurological effects. Still, given the magnitude and complexity of the brain function, further studies of the effect of intraoperative hypothermia in hippocampus-related memory should be given a high priority.

## 5. Conclusions

The CA1 region contains abundant blood vessels and is susceptible to ischemia. Intraoperative hypothermia disturbed the blood supply by causing the dysfunction of VECs and the phenotypic transformation of VSMCs. Ultimately, neurons in the CA1 region displayed morphological pyknosis and biological suppression of Arc. The findings from this study suggest future research directions. Further measurement of regional blood flow in the hippocampus might allow characterization of vascular function during intraoperative hypothermia. Aside from neuronal morphological pyknosis and reduced Arc expression, subcellular synaptic plasticity might also be an important neuroelectrophysiological change. Lastly, but importantly, intraoperative hypothermia is very common, but usually neglected. While intraoperative hypothermia can be prevented by thermal insulation, it cannot be completely eliminated. Notably, some operations require hypothermia. The recovery of memory function is important to determine the quality of life after such operations. This study could provide a novel understanding of the effect of intraoperative hypothermia in the hippocampus. As such, it might identify a new research target and treatment strategy for memory decline induced by intraoperative hypothermia.

## Figures and Tables

**Figure 1 brainsci-12-00692-f001:**
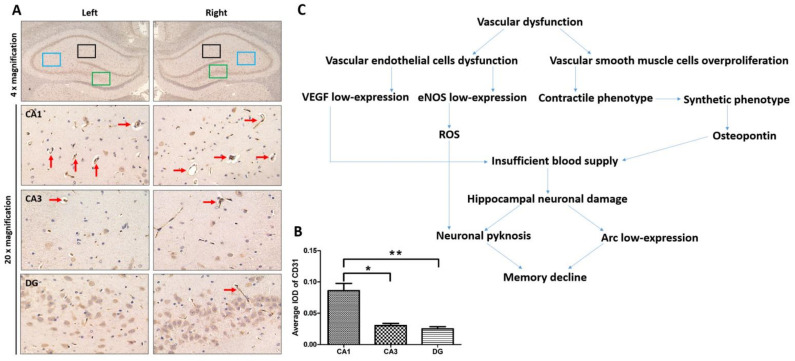
Analysis of hippocampal blood vessels and the relationship between vascular dysfunction and neuronal damage. (**A**) CD31 immunohistochemical staining of blood vessels (red arrows) in the rat bilateral hippocampus. In 4× magnification images, the circled rectangles in the different regions of the hippocampus were magnified by 20×. Black rectangles represent CA1, blue rectangles represent CA3, and green rectangles represent DG. (**B**) Integrated optical density (IOD) values of CD31. * *p* = 0.0027 vs. CA3, ** *p* = 0.0019 vs. DG. (**C**) The schematic diagram demonstrates the transition from vascular dysfunction to neuronal damage in the study. Statistics data were presented in Table 1.

**Figure 2 brainsci-12-00692-f002:**
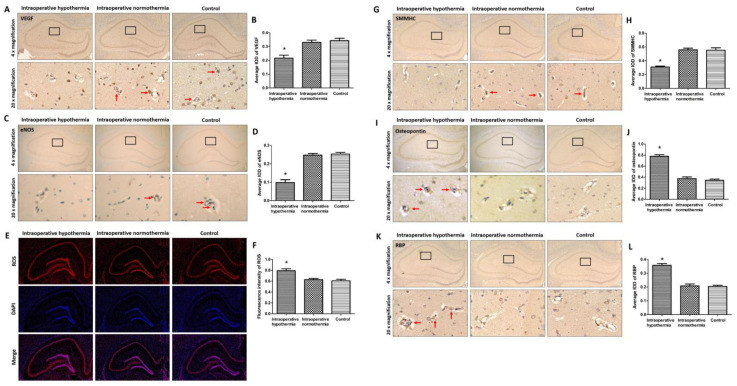
Intraoperative hypothermia leads to endothelial dysfunction and the phenotypic transformation of vascular smooth muscle cells. (**A**–**D**) Immunohistochemical staining of vascular endothelial growth factor (VEGF) and endothelial nitric oxide synthase (eNOS) in rat hippocampus and summary integrated optical density (IOD) values. * *p* = 0.0051 vs. Control, in quantification of VEGF; * *p* = 0.0001 vs. Control, in quantification of eNOS; * *p* = 0.0077 vs. Control, in quantification of ROS. (**E**,**F**) Reactive oxygen species (ROS) fluorescence staining of rat hippocampus and fluorescence intensity (4× magnification). (**G**–**L**) Immunohistochemical staining of smooth muscle myosin heavy chain (SMMHC), osteopontin, and retinol binding protein (RBP) in rat hippocampus and summary IOD values. * *p* = 0.0007 vs. Control, in quantification of SMMHC; * *p* < 0.0001 vs. Control, in quantification of Osteopontin; * *p* = 0.0001 vs. Control, in quantification of RBP. In 4× magnification images, black rectangles in the CA1 region are selected to perform 20× magnification. Intraoperative hypothermia: rats experienced hypothermia during surgery. Intraoperative normothermia: rats maintained normal body temperature during surgery. Control: rats were fed normally without operation. Statistics data were presented in Table 1.

**Figure 3 brainsci-12-00692-f003:**
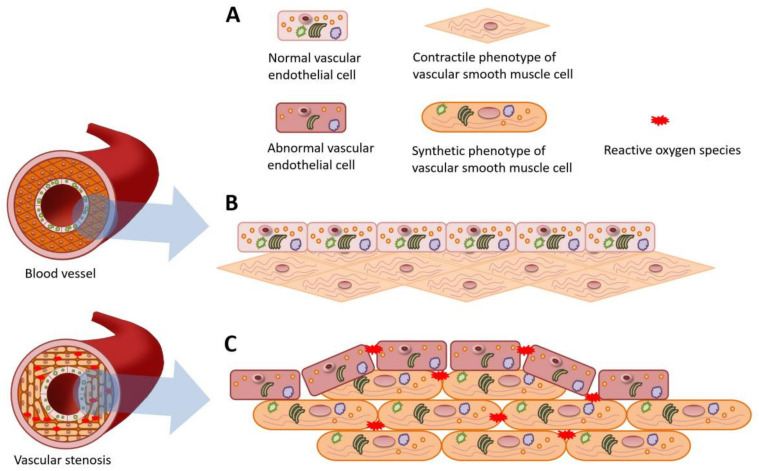
Schematic representation of the effects of intraoperative hypothermia on vascular endothelial cells (VECs) and vascular smooth muscle cells (VSMCs). (**A**) Key legends for the schematic. (**B**) Normal VECs and VSMCs with contractile phenotype make up the wall of blood vessels, which preserve the blood supply. (**C**) Dysfunctional VECs increase the vascular permeability while VSMCs with synthetic phenotype proliferate excessively, causing stenosis of the vascular lumen.

**Figure 4 brainsci-12-00692-f004:**
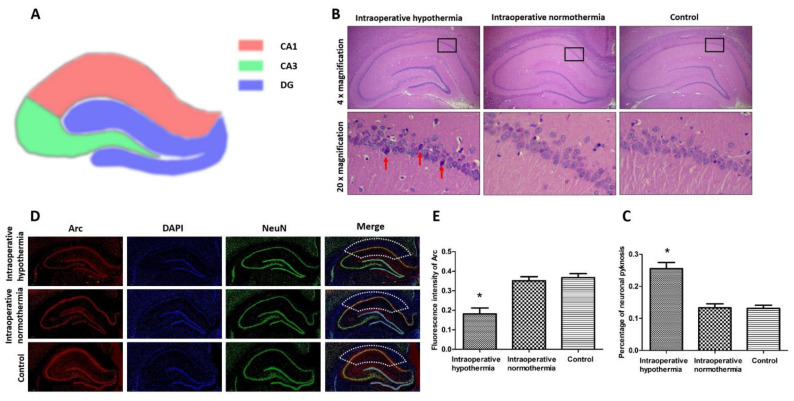
Histopathology of the hippocampus caused by intraoperative hypothermia. (**A**) Cross-sectional diagram of rat unilateral hippocampus with three regions encoded by different colors. (**B**,**C**) Neuronal pyknosis (red arrows) detected by H&E staining and summary results. Black rectangles in the CA1 region are selected to perform 20× magnification. * *p* = 0.0015 vs. Control. (**D**,**E**) Representative immunofluorescence (4× magnification) staining for Arc in the hippocampus. The red fluorescence shows Arc expression. The blue fluorescence represents nuclei stained with 4′,6-diamidino-2-phenylindole (DAPI). The green fluorescence shows neurons labeled with NeuN. In the merged figure, the CA1 region is framed in a dotted line. * *p* = 0.0026 vs. Control. Intraoperative hypothermia: rats experienced hypothermia during surgery. Intraoperative normothermia: rats maintained normal body temperature during surgery. Control: rats were fed normally without operation. Statistics data were presented in Table 1.

**Table 1 brainsci-12-00692-t001:** ANOVA analysis of factors associated with vascular and neuronal function.

Effect	R-Squared	F	*p*
CD31	0.9478	36.30	0.0027
VEGF	0.8281	14.45	0.0051
eNOS	0.9524	59.98	0.0001
ROS	0.8021	12.16	0.0077
SMMHC	0.9100	30.33	0.0007
Osteopontin	0.9632	78.55	<0.0001
RBP	0.9472	53.85	0.0001
Neuronal pyknosis	0.8866	23.45	0.0015
Fluorescence intensity	0.8617	18.69	0.0026

VEGF, vascular endothelial growth factor; eNOS, endothelial nitric oxide synthase; ROS, reactive oxygen species; SMMHC, smooth muscle myosin heavy chain; RBP, retinol binding protein.

## Data Availability

All data included in this study are available upon request from the corresponding authors.

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
