# Peer review of "Intraoperative Hypothermia Induces Vascular Dysfunction in the CA1 Region of Rat Hippocampus"

_brainsci, 2022, doi:10.3390/brainsci12060692_

Round 1

Reviewer 1 Report

The manuscript entitled “Intraoperative hypothermia induces vascular dysfunction in the CA1 region of rat hippocampus” by Li et al investigated intraoperative hypothermia causes vascular dysfunction in the hippocampus that is associated with neuronal dysfunction and cognitive impairment. The topic of the manuscript is interesting, however, there are major concerns discussed below:

Major:

The diameter of adult rat cerebral microvessels varies depending on whether the vessels are capillaries, precapillary arterioles, or penetrating arterioles. Most of the capillaries have a size of ~ 5-10 um and the other microvessels are larger than 10 um. The current manuscript investigated the vascular density in the hippocampus region which are microvessels. However, the current study that uses 5 um sections to investigate microvessels density is not acceptable. Since it will only capture a small piece or a fragment of microvessels making it is impossible to rigorously compare the vascular density between groups. A thicker brain section around 50-60 um should be used to include the whole microvessels segment. This also applies to the eNOS, VEGF, and smooth muscle phenotype markers staining. In addition, there is no description of how the data were quantified and how many sections were analyzed. A series of bran sections should be processed and quantified to reflect the whole vascular structures in the hippocampus region.

Vascular smooth muscle phenotypic change from contractile to synthetic can induce the proliferation of smooth muscle cells leading to increased artery size. This phenomenon always happens in big arteries such as the middle cerebral artery in the brain. However, the hippocampus microvessels especially capillaries that comprise a vast majority of hippocampus vasculature lack smooth muscle cells. It is unclear how the smooth muscle cell phenotypic changes in this context correlated with hippocampus vascular dysfunction and how these contribute to neuronal dysfunction. Also, an increase in artery size can cause elevated cerebral blood flow which contradicts the hypothesis of the current manuscript. Again, with current results from 5 um sections staining no conclusion can be reached that there are phenotype changes of SMCs. 

What are the effects of intraoperative hypothermia on pericytes?

There are no neurobehavior tests in the current manuscript showing whether the intraoperative hypothermia results in cognitive impairment.

Minor:

Figure 1c, the results in the current study cannot support this conclusion such as there is no evidence in the current study proving ROS can induce SMC phenotype changes.

Please make sure that the references are properly cited. eg: there should be a reference for “Recently, studies showed that there were no obvious brain abnormalities related to synaptic plasticity, neurotransmitters, or other physiological function. Additionally, brain tissue is sensitive to ischemia” in the introduction.

Reviewer 2 Report

The manuscript is well written and presents results that are compatible with the authors' conclusions. However, just a few modifications are needed.

-All animals were used to all techniques? I suggest the authors to provide this information about the nunber of animals per analysis in a more clearly way.

- The statistical results are not provided. It is really important to present them.

-How many images and sections per animal were used in the analysis?

- I really liked the way the results were expressed, but it felt a little repetitive with the discussion sometimes. And, again, the statistical results must be provided.

-The signal to show that one group is not different from the control is not needed. I understand that the authors wanted to evidenced this results, but when we look to a graphic for the first time, we automatically associated a sign to something different.

Reviewer 3 Report

The manuscript by Tianjia Li et, al. investigates the effects of intraoperative hypothermia on the vascular function in the CA1 region of hippocampus in a rodent model. Given that the vascular dysfunction has a huge impact on the cognitive impairment, investigating the effect of hypothermia on the hippocampus function is very interesting. However, there are several weaknesses that need to be addressed:

Major comments:

  1. The authors investigated the vascular dysfunction by assessing the function of VECs and phenotypes of VSMCs. It is also described at Line 252 that “The vascular wall consists of VECs and VSMCs…”. However, the very end unit of the cerebral vascular structure should be the endothelial cells and surrounding pericytes, which are the components of the blood brain barrier (BBB) and the neurovascular unit (NVU), but not the SMCs. It should be clarified if the authors are studying the arterioles or the capillary vessels, and the impact of the hypothermia on the BBB and NVU should be discussed.
  2. The study was designed to have the intraoperative hypothermia and observe the vascular and neuronal damage in CA1 region. However, before the animals were sacrificed at 12 days after surgery, why there is no behavior tests to support the hypothesis?
  3. The investigators found the accumulation of ROS in the hippocampus in the histological study. However, previous studies have demonstrated that the hypothermia can reduces calcium loading and mitochondrial transition, and maintain cellular energy which all can scavenger ROS. The mechanisms of how hypothermia increased the ROS accumulation in the current study need to be proved with the quantitative studies to support the conclusion. The difference between previous therapeutical hypothermia studies and the current study also need to be discussed.
  4. The authors suggested the hypothermia caused the cerebral hypoperfusion. However, there are no experiments and results, such as cerebral blood flow measurement, to support this point of view.
  5. Line 146 “Compared with other regions, there are more blood vessels in the CA1 region, which demonstrates its susceptibility to ischemia”, how so? Isn’t the region with less vessels more susceptible to ischemia?

Minor comments:

  1. There is no clear hypothesis in the introduction part. The last paragraph in Introduction could be moved to discussion.
  2. The histological study is the only method used to evaluate all the conclusive results, larger and high-definition images are necessary.

Round 2

Reviewer 1 Report

Thanks for the author responding to the comments. It can be achieved by confocal microscope with z-projection for imaging thick sections and comparing the staining differences. All of the data in the current manuscript largely depends on immunohistochemistry or immunostaining. There is no expression assay et al. western-blot to validate the observation in the immunohistochemistry or immunostaining studies. Again, the fluorescence intensity can be varied largely across different segments of the vessel wall. The reference (DOI:10.1038/s41586-021-04369-3 ) provided in the letter of response is not relevant at all to the current study, the 10 um sections were not used for the quantitation of protein expression levels. 

Please specify which segment of the arteriole was measured resulting in a diameter of around 21.16±0.87μm and explain in detail how the arteriole sections were compared at the same level. The penetrating arteriole diameter varies at different depths of the brain. How does the author distinguish arteriole and venous? Please note that most of the neurovascular coupling is at the capillary level. Neurovascular coupling is the interaction between endothelial cells, pericytes, and astrocytes. The author mentioned "arteriole-related blood supply", how the results in the current manuscript demonstrate that there is smooth muscle cell and arteriole dysfunction leading to damaged neurovascular coupling?. Please provide cerebral blood flow measurement data to support the conclusion of cerebral hypoperfusion. 

The manuscript can not draw the conclusion that ROS causes SMCs phenotypic change just by citing other people's manuscripts. All of the experimental models and conditions were different. And only using one marker for synthetic and contractile is not sufficient to make the conclusion. How about Lamp1, Lamp2, and SMA.

Reviewer 3 Report

The introduction part could be improved by giving a succinct writing to help the readers grasp the main idea of the study. Other than that, the authors have addressed all the previous comments and provided a much improved manuscript.
